# The Sound of Vision Project: On the Feasibility of an Audio-Haptic Representation of the Environment, for the Visually Impaired

**DOI:** 10.3390/brainsci6030020

**Published:** 2016-06-27

**Authors:** Ómar I. Jóhannesson, Oana Balan, Runar Unnthorsson, Alin Moldoveanu, Árni Kristjánsson

**Affiliations:** 1Laboratory of Visual Perception and Visuo-motor control, Faculty of Psychology, School of Health Sciences, University of Iceland, Reykjavik 101, Iceland; ak@hi.is; 2Faculty of Automatic Control and Computers, Computer Science and Engineering Department, University Politehnica of Bucharest, Bucharest 060042, Romania; oana.balan@cs.pub.ro (O.B.); alin.moldoveanu@cs.pub.ro (A.M.); 3Faculty of Industrial Engineering, Mechanical Engineering and Computer Science, School of Engineering and Natural Sciences, University of Iceland, Reykjavik 101, Iceland; runson@hi.is

**Keywords:** visually impaired people, brain plasticity, adaptation, sensory substitution, training

## Abstract

The Sound of Vision project involves developing a sensory substitution device that is aimed at creating and conveying a rich auditory representation of the surrounding environment to the visually impaired. However, the feasibility of such an approach is strongly constrained by neural flexibility, possibilities of sensory substitution and adaptation to changed sensory input. We review evidence for such flexibility from various perspectives. We discuss neuroplasticity of the adult brain with an emphasis on functional changes in the visually impaired compared to sighted people. We discuss effects of adaptation on brain activity, in particular short-term and long-term effects of repeated exposure to particular stimuli. We then discuss evidence for sensory substitution such as Sound of Vision involves, while finally discussing evidence for adaptation to changes in the auditory environment. We conclude that sensory substitution enterprises such as Sound of Vision are quite feasible in light of the available evidence, which is encouraging regarding such projects.

## 1. Introduction

In the late 1980s, computers with an attached “mouse” started to become household objects. The mouse translated a motor movement on a horizontal surface (a desk) into the movement of a visual stimulus (a pointer) on a vertically oriented object (a screen). This involved two complicated transforms that the user’s brain had to interpret: from motor movement to visual perception and from a horizontal plane to a vertical one. This was neither easy nor intuitive, but computer users quickly mastered it, learning to use the mouse with great accuracy. Today this is an effortless activity that virtually everyone has mastered and most recently modern information technology has transferred this skill to touchscreen devices. What this demonstrates is the brain’s capacity to recalibrate its operative characteristics to meet new demands.

Such flexibility is at the heart of the current review where the aim is to discuss research relevant to the question of the feasibility of the Sound of Vision project. The aim with Sound of Vision [1] is to implement and validate original non-invasive hardware and software aimed at assisting visually impaired people by creating and conveying an auditory and haptic representation of the surrounding environment—in as much detail as possible. This representation will be created, updated and delivered to the visually impaired continuously and in real time. Sound of Vision will help the visually impaired in any kind of environment (indoor/outdoor), without the need for predefined tags/sensors located in the surroundings. The current aim is not only to aid with obstacle avoidance etc. but to provide as rich a picture of the environment as possible, including object identification, the distance and direction to stimuli, and their height and width, to as large an extent as possible—a picture that comes as close to conveying a natural sense of vision as possible. The information about the environment will be acquired through video cameras and the relevant information extracted from each frame and converted to auditory and haptic feedback. The acquired feedback will subsequently be conveyed to the user through both audition and haptics, where the aim is to strategically utilize strengths and avoid weaknesses of each modality for this purpose. The auditory feedback will be played to the user through specially designed headphones that do not block natural environmental sounds. To perceive the haptic feedback, users will wear a belt containing vibrators that will be turned on and off in accordance with the information conveyed to the user. Such sensory substitution involves remapping of sensory input, perceptual interpretation and motor response just as in the example of the computer mouse.

An obvious prerequisite for the usefulness of such a system is that the brain will be capable of using such information and recalibrating its spatial navigation systems so that such real time information can be utilized. We therefore present a review of a number of issues that are important for the creation of such a sensory substitution device. Our main emphasis will be on functional plasticity in the brain from the perspective of non-visual sensory substitution and spatial perception. In the first section some examples of neuroplasticity of the adult brain are reviewed with an emphasis on functional changes in visually impaired people (VIP) compared to sighted people. In section 2, we discuss effects of *adaptation*, specifically on long-term and short-term effects of repeated exposure to particular stimuli on brain activity (see Table 1 for overview). Such adaptation is key to enabling users to adjust to the sensory substitution devices. In section 3 we discuss sensory substitution such as when a region devoted to a particular function takes on another function (see Table 2 for overview). In section 4 we discuss research on adaptation to changes in auditory environments (see Table 3 for overview). In section 5 we review evidence pertaining to the training of sound localization (see Table 4 for overview). Finally, we present concluding remarks on our review of neuroplasticity, adaptation and sensory substitution and the implications for the feasibility of an enterprise such as Sound of Vision. All of these are crucial questions that must be answered before any serious attempt is made at providing sensory substitution for impaired sensory systems. Since the Sound of Vision project involves providing feedback both through haptics and audition we focus on studies relevant to sensory substitution for vision through both these modalities. Note that even if our focus is on the Sound of Vision project, the general principles that we uncover most likely apply to sensory substitution devices in general.

To preview our conclusions, our review indicates that such an undertaking is feasible given the changes in brain function that can occur even in relatively rigid adult brains. This furthermore suggests that with time, the mapping of the provided information to spatial codes may involve less and less effort, so that strained connections can over time become effortless just as in the example involving the computer mouse.

The newborn brain contains the vast majority of the neurons it will ever contain but its size is only about 25% of the size of the adult brain. Brain size increases rapidly at young ages through increased interconnections (i.e. synaptogenesis) between neurons and increased myelination of the neuronal axons [2,3,4]. After birth, experience strongly influences the development of the brain [5,6] and it has been known for decades that sensory neurons do not develop normally if they do not receive the required stimulation (e.g. [7,8]). A longstanding view was that the functionality of the adult brain is mostly fixed. Recent evidence disputes this view, however, and considerable plasticity has been observed in adult brains [9], and, importantly for the current project, this also applies to vision [10,11]. The brain does indeed have considerable abilities for adapting its operative characteristics to changed circumstances. Note that the current review focuses on such neural adaptation, not on devices for sensory substitution. For this we refer the reader to an excellent review by Elli, et al., [12].

## 2. Evidence for Structural and Functional Differences in Auditory and Haptic Perception between Visually Impaired and Sighted People

Brain structure in congenitally blind (CB), late blind (LB) and sighted people differs. In some areas the volume and the surface is smaller in CBs and LBs than in sighted controls (SCs) and some evidence suggests that both CBs and LBs have reduced grey matter compared to SCs [13]. The surface of the primary and associative visual cortices is smaller in CBs than SCs and the visual cortex in CBs has been found to be significantly thicker than in LBs [13]. This suggests that the structural differences might be associated with the lack of vision during development but not with disuse-driven atrophy or adult compensation. Interestingly, Park et al. [13] found that the cortex was thicker in CBs, than in SCs, in some areas that process visual information in sighted people (e.g., the frontal eye field and the anterior cingulate cortex). Furthermore, Park et al. [13] found a significant decrease in cortical thickness in auditory brain areas in CBs and proposed that such thinning is reflected in better auditory performance. Park et al. [13] also found significant thinning in the somatosensory cortices of CBs, which have high functional connectivity with the visual cortex and suggest that this may reflect greater tactile skills by the CBs. All these morphological alterations in CBs suggest that reorganization of visual cortex takes place in conjunction with other sensory cortices.

In light of the structural differences between visually impaired and sighted people discussed above, it is reasonable to expect differences in their performance on auditory and tactile tasks, compared to participants with normal vision. In a clever experiment, Stevens and Weaver [14] demonstrated that the performance of visually impaired participants on temporal order judgments (TOJ), and auditory backward masking (ABM) tasks was better than of sighted people. In the TOJ-task two tones of differing frequency were played with different stimulus onset asynchronies (SOA; using the staircase method) and the task was to judge which sound was played first. The purpose of the TOJ-task was twofold: to compare TOJ performance between early blind (EB: congenitally blind or people that became blind in their first 2 years of life) and sighted blindfolded (SB) and sighted non-blindfolded (SC) participants and find the minimal SOA between the two tones where each observer could still tell them apart. The individual’s SOA was subsequently used in the ABM experiment to eliminate individual differences. The difference in mean SOA threshold between EB and SC groups was significant but other differences were not. The task in the ABM experiment was to judge if the order of two tones (high or low) was the same or different. A mask followed at varied delays, but not on all trials. Interestingly, the mask delay did not affect the performance of the EBs but the SCs and SBs needed on average a 160 milliseconds (ms) delay to reach the same performance level as on no-masking trials [14], which indicates that the EB processed the auditory stimuli faster in both the TOJ and the ABM experiments.

Some studies have revealed better auditory spatial localization in blind than sighted individuals [16,21,22,76]. However, there is no consensus regarding the mechanisms for such enhancement. Sound localization is affected by eye-movements, even for congenitally blind people. In Després et al. [15] (experiment I), sounds were played from loudspeakers located 5°, 10°, 15° and 20° to the left and right of participants’ mid-sagittal plane. Three conditions were compared: (i) no eye movements allowed; (ii) observers were to look in the direction of the sound; (iii) observers were to look in the opposite direction to the sound. In all conditions the participants had to point to the direction of the sound. The performance of the CB and SC groups was similar. The directional pointing error was lowest when observers had to look and point in the same direction (≈ 8°), higher when they had to look in one direction and point into the other (≈ 11°) and highest when no eye movements were allowed (≈ 15°) [15]. These results underline the importance of gaze in spatial localization and are consistent with studies that have not found better performance for VIPs in locating straight ahead sound sources. In experiment II [15], where the task was to judge whether a sound came from 10° above or below a horizontal plane level with the ears, response times of the CB group were shorter than of the SC group when the sounds were presented from behind (± 20° from the dorsal-ventral center line) but not when the sounds were presented from ahead with similar deviation from the dorsal-ventral center line [15]. When judging the direction of a sound source in the horizontal plane, interaural time difference is an important cue which is absent in the vertical plane. We might therefore expect that spectral cues—and vision—are more important for the vertical than the horizontal planes and when locating a sound source monaurally. Voss et al. [17] compared the performance of EBs and SCs in binaurally locating sounds in the vertical plane and monaurally and binaurally in the horizontal plane. There was no difference in the binaural horizontal tasks but the results from the monaural task suggest that the EBs can be divided into those who perform similarly to SCs and those who perform better than SCs. But this comes at a cost. When locating sound sources in the vertical plane the performance of the superior group was significantly worse than of the SC [17]. Furthermore, Finochietti et al. [18] show that although both VIPs and SCs show similar performance in detecting the direction of a moving sound, the performance of EBs in locating the end point of the moving sound is worse than for both LBs and SCs. Those results further highlight the importance of vision for sound localization. Overall it seems that better performance in one aspect might come at a cost in performance in other aspects (see e.g., [77], for review).

Röder et al. [16] presented auditory stimuli using 8 speakers located at: 0°, 6°, 12°, 18° (central speakers array) and 90°, 84°, 78° and 72° (peripheral speakers array) from the observers straight ahead. A target sound (probability = 0.16) and a standard sound (probability = 0.84) were presented. Both sounds could appear from any speaker. On half the runs observers were instructed to respond, by button-press only when the target sound was presented from the 0° speaker (central condition); while in the other half they responded only when the sound came from the speaker 90° away (peripheral condition). In both cases, the target could, however, appear from any of the remaining 3 speakers from the array corresponding to each condition, but responding to them constituted an error. No response was required for the standard sound. In the central condition the SCs responded more often correctly to the 0° speaker than the CBs and the error rates declined equally fast within both groups towards the other 3 speakers. In the peripheral condition, correct responses to the 90° speaker were similar, but the error rates declined faster within the CB group than the SC group. Attention effects (attended minus unattended amplitudes) were measured with tomographic voltage maps and the N1 values (negative-going evoked potential appearing 100 ms after stimulus presentation) in the central condition did not differ between groups but their amplitude declined faster in the central than peripheral conditions. In the peripheral condition the amplitude of N1 decreased faster for the CBs. This indicates that CBs process peripheral auditory stimuli more efficiently than sighted people, but sighted people are more accurate in responding to central auditory stimuli.

There is no consensus that VIPs respond faster or more accurately to auditory stimuli than sighted people. Fieger et al. [19] found, using the same procedure as Röder et al. [16], that LBs responded more slowly than sighted people when responding to straight ahead and peripheral sounds. Although it took the LBs longer to respond to the auditory stimuli, the results from Fieger et al. [19] suggest that LBs processed peripheral stimuli more efficiently than the SCs. N1 amplitude (at Cz) did not differ between groups neither when responding to central nor peripheral stimuli. When attending to the central speakers there were no differences in the gradient of P3 (at IPz) between the groups, but when attending to peripheral speakers the decline in P3 amplitude with distance from the attended speaker was significant for LBs but not SCs. Fieger et al. [19] concluded that both CBs and LBs process peripheral auditory stimuli more efficiently than sighted people while using differing mechanisms. In a more recent study Lerens and Renier [20] found that EBs responded faster to auditory stimuli than SCs irrespective of whether the stimuli were presented from straight ahead or from the periphery (± 90° from center). The early blind participants in Lerens and Renier’s study were either congenitally blind or became blind within the first 3 years of life. The difference between the results of Lerens and Renier [20] and Fieger et al. [19] may be a consequence of when the participants became blind. Furthermore, the level of blindness might also affect sound localization performance of VIPs. In an earlier study, Lessard et al. [21] found no differences in sound localization performance between congenitally blind and sighted participants but the performance of participants with residual sight was inferior to the other groups. Further support for similar performance of VIPs and SCs in sound localization tasks comes from Voss et al. [22,23]. In Voss et al. [22] the task was to locate a sound source but in another study conducted by Voss et al. [23] the task was to compare locations of sounds. No significant differences between EBs and SCs were found in either experiment. Interestingly, though, in Voss et al. [22] the ability to discriminate between distances was also examined. Voss et al. [22] found that the ability of early and late blind to judge different distances was not significantly different but better than the performance of the sighted controls.

Gori et al., [24] compared performance as a function of distance between sound sources, presenting three sounds to the participants through loudspeakers ranging ± 25° from their straight ahead. The task was to judge whether the middle sound was closer to the first (e.g., left) or the third (e.g., right). The CBs performed at chance level while the SCs solved this task well. These results might seem to be add odds with previous results but the inferior ability of the CBs, compared to the SCs might be a consequence of methodological differences between this and the other studies [78]. Gori et al. [24] also tested the ability to compare the distances of sounds using similar methods as in the spatial test except that all sounds were played from the central speaker, finding no significant differences between the groups. When the task was to tell which of two successively presented sound were more to the left (or right), the CBs performed similarly to the SCs.

Röder et al. [25] measured response times to acoustic oddball targets and recorded event-related potentials (ERPs) from frontal, central, temporal, parietal and occipital scalp positions against the right mastoid for 11 CBs and 11 SCs for comparison. They found that CBs responded significantly faster to acoustic stimuli than SCs. Furthermore, the peak amplitude of N1 at the temporal and parieto-occipital electrodes was larger and recovered faster in CBs than in SCs. Röder et al. [25] concluded that the same brain areas process elementary auditory stimuli for CBs than SCs, but more efficiently for the former.

It is more important for VIPs to be able to discriminate between various acoustic and haptic information than sighted people. In a crossmodal experiment (see e.g., [79] and [80] for review), Occelli et al. [26] used acoustic and haptic stimuli, investigating participant’s ability to tell which stimulus was presented first. The acoustic stimulus could appear before or after the haptic stimulus. Both stimuli could appear on the same side or one on the right and the other on the left. The mean JND was the same for SCs whether the stimuli appeared at the same location or not. But for the VIPs the JND was lower when the stimuli appeared at different locations than at the same location. There were no differences between early- and late-blind participants.

Röder et al. [27] found effects of early vs. late blindness on the ability to distinguish the temporal order of haptic stimuli. Tactile stimuli were presented to the left or right middle finger with different stimulus onset asynchronies. On half the trials participants were tested with crossed arms but uncrossed on the other half. Of the 25 SCs, 13 were blindfolded while the other 12 were not. The performance of the SCs did not depend upon whether they were blindfolded or not but their JNDs were significantly higher when they responded with crossed, than with uncrossed arms. The visually impaired (VIP) group consisted of 10 CBs (congenitally blind) and 5 LBs (late blind) participants and all were proficient Braille readers. Their performance was not significantly worse in the crossed condition than in the uncrossed condition, but the performance of the LBs was indistinguishable from the performance of SCs in the crossed arms condition. In both the crossed and uncrossed conditions the performance of CBs was better than that of the SCs [27].

Visual and haptic illusions frequently provide important information about perception. In Hötting and Röder [28] a tactile stimulus was presented to participant’s right index finger. On each trial one to four 50 ms tactile stimuli were presented with an ISI of 200 ms. The auditory stimuli (duration = 10 ms, ISI of 100 ms) were presented through 2 loudspeakers. The number (0 to 4) of the auditory stimuli varied between trials and the first preceded the first tactile stimulus by 25 ms. Participants judged how often they perceived the tactile stimulus. In all groups the perception of 1 tactile stimulus was affected by the number of auditory stimuli but the effect differed between groups (15 CB, all proficient Braille readers; 15 sighted blindfolded (SB) and 15 sighted not blindfolded (SC) observers). All participants reported significantly more tactile stimuli when 2 tones, rather than 1 or none were presented. For the SC group the illusion was also reliable when 3 or 4 tones were presented but not for the SB group. For the CB group the illusory effect significantly decreased when 3 or 4 tones were presented compared to when 2 tones were presented [28].

Hötting et al. [29,30] studied the effects of unimodal, crossmodal and intermodal attention on ERP patterns and behavioral responses of 16 sighted [29] and 15 visually impaired [30] participants. The task was to attend to one modality at one spatial position at a time and respond. Accuracy was similar in both experiments. The CBs responded significantly faster to tactile stimuli than the SCs but the difference was not significant for auditory stimuli. The ERP data suggest that the selection process differs between CBs and SCs. ERP studies frequently show that the negativity observed 200 ms following the presentation of deviant auditory or somatosensory stimuli is more posteriorly distributed for VIPs than SCs. In the earliest time course (<200 ms after stimulus presentation) no reliable topographical differences between CBs and SCs were found and Hötting et al. [29,30] therefore proposed that the initial selection process involves similar brain areas in CBs and SCs. The selection mechanism might however differ when stimuli have to be filtered by two factors (location and modality). Hötting et al. [29,30] suggest that the initial selection by SC is based on both modality and location but the CBs filter mainly by modality and that at later stages the CB suppress processing of task irrelevant stimuli, while SCs amplify processing of task relevant stimuli (attended modality and location). Hötting, et al., [30] therefore suggest that multisensory processes are experience dependent and that the lack of vision seems to affect auditory and tactile interactions.

Wan, et al. [31] studied pitch and pitch-timbre discrimination in 11 congenitally blind, 11 early blind (between 1.4 and 14 years of age), 11 late blind and 33 sighted people. The sample was matched on age, gender, nature and extent of musical training, current musical activity, and whether they had perfect pitch. When accuracy was compared over difficulty levels, accuracy of CBs was better than of their matched controls. When the accuracy of the EB was compared to the accuracy of their matched controls the only significant difference was seen in the easiest version of the task. In the pitch-timbre discrimination task, CBs and EBs were better at detecting changes in pitch and timbre than their matched sighted participants. In both the pitch and pitch-timbre discrimination tasks, performance of CBs and EBs was significantly better than of their matched SCs but no significant difference in performance was found between LBs and their matched controls [31]. Wan et al. [31] concluded that becoming blind at a young age enhanced auditory acuity.

There is some evidence for inferior performance of early-blind people in object identification accuracy compared to sighted or late blind people but also that EBs might be faster at identification (see e.g., [32]). Indeed, Postma et al. [32] found that both EBs and LBs were faster than sighted blindfolded people (SB) in putting geometric shapes in their correct places on a wooden board. After rotating the board 90° the VIPs were still faster than the SBs [32]. Withagen et al. [33] studied both adults and children (matched by age and gender) where the task was to compare 4 artificial objects in 16 groups by handling them. Sighted adults were slower than VIPs in solving the task.

There is evidence that the visually impaired have enhanced tactile acuity. Goldreich and Kanics [34] compared tactile acuity of 43 VIPs and 47 SCs. The VIPs were split into different groups according to their level of impairment, experience of being visually impaired and experience with Braille. Performance of the VIP group was significantly better than of the SCs. When impairment level, Braille reading and experience of being visually impaired were taken into account, the superiority of the VIPs compared to SCs was still significant in all groups [34]. Additionally, in a similar study [81] with more fingers tested, the performance of the VIPs was significantly better irrespective of whether the finger tested was their preferred Braille reading finger or not.

## 3. Neural Adaptation

Exposure to certain stimuli can alter subsequent responses to them, a process called *Adaptation*. The altered responses are consequences of changes in neuronal response properties (see e.g., [82]). Questions regarding adaptation to stimulation are highly relevant to sensory substitution enterprises since changes of neural operation following prolonged stimulation can provide clues about observers’ responses to SS devices.

Neural adaptation has been observed both with single-cell physiology and non-invasive functional imaging. One example is when a stimulus with the same properties (e.g., color and/or shape) is presented repeatedly, RTs towards that stimulus decrease (priming, [35,36]. The same holds if the stimulus is presented repeatedly at the same location [35,37]. Kristjansson et al. [39] used a “pop-out” visual search paradigm to investigate the effect of repetition of stimulus properties on RTs and blood oxygen level-dependent (BOLD) signal, measured with fMRI. In the pop-out paradigm the target differs from distractors on a single feature and can therefore easily be found, i.e., it pops out. Many neuroimaging studies have found that with repeated presentation BOLD activity decreases (repetition suppression, [38,40,41,42,83]. Consistent with other studies, Kristjansson [39] observed BOLD repetition suppression in several brain areas, in some areas when color was repeated, in others when location was repeated and still others when both color and location were repeated (see also [84,85]. The repetition effects last relatively long and do not necessarily include any structural or functional changes in the brain and are therefore better referred to as adaptation. Kristjánsson et al. [39] hypothesized that these effects of repeated presentation reflect lesser effort while processing repeated stimuli.

Such activity suppression has been seen in single-cell recording as well. Bichot and Schall [51,52] observed that single-unit responses to distractors in the frontal eye fields of Macaque monkeys were decreased by target priming in a visual search paradigm similar to the one tested by Kristjánsson et al. [39], indicating that priming of pop-out may cause a selective “pruning” of the FEF population response to a given search display (see e.g., [43]).

EEG measures have also been used to measure adaptation. The psychometric properties of EEG have been evaluated and resulted in very good internal consistency, split-half reliability, reliability, and test-retest stability [86]. Adaptation, in general, reduces the EEG signal [44,45]. When investigating whether selection processes depend more on color or form, Rentzeperis et al. [46] compared behavior and EEG results on responses to radial and concentric colored Glass patterns (see e.g., [87]). The behavioral experiment resulted in stronger adaptation to form than color that was echoed in the EEG results. The amplitude of the N1 component was significantly lower when color was repeated than when color was not repeated. Later (>300 ms) form repetition effects became significant. These results show that EEG measures can be reliable indicators of neural adaptation.

Similar adaptation occurs for faces. Rossion and Boremanse, [47] showed that when the same face was repeatedly presented, the amplitude measured over the occipito-temporal scalp decreased significantly compared to when the face on the current trial differs from the previous one, i.e., neuronal activity decreased so no changes between trials occur. Interestingly, when the faces were inverted the adaptation effect disappeared, suggesting that the upside-down faces looked mostly the same whether or not they were the same or different [47]. In an experiment with a similar procedure Gerlicher et al. [48] investigated whether different facial expressions (e.g., neutral, fearful, happy) modulated adaptation. Gerlicher et al. [48] found that the adaptation effects were only significant in the neutral condition. The lack of adaptation in the fearful condition is in accordance with previously observed effects of threatening facial expressions on latency, i.e., people take longer to respond to stimuli in trials where threatening than neutral faces are involved, as has been shown in various studies (for discussion of some of those studies see e.g., [88].

It is important to connect these short-term changes to longer-term changes in neural activity, and there is indeed evidence that such adaptation-related reductions in neural activity also occur in the long-term and are accompanied by longer-term changes [89]. Repetition suppression may indeed reflect facilitation of perceptual identification [90] and lead to faster or more efficient processing in a number of brain regions [91]. In Dobbins et al. [49], participants judged whether or not an everyday object was bigger than a shoebox. Faster RTs were found, for previously presented versus novel stimuli, in conjunction with repetition suppression in a number of regions. In Wander et al. [50] observers tried to move a cursor through brain-computer interfacing (BCI) up or down while it moved over the screen and to hit a designated target at the other side of the screen, Wander et al. [50] found that performance improved significantly with increased trial number. More interestingly, activity as measured with ECoG (implanted electrodes) in a number of brain areas decreased as proficiency in the task improved, consistent with the idea that repetition suppression is connected with longer-term changes in neural activity. As task proficiency improves, the task requires less effort, and this is reflected in the results from Wander et al. [50], which is consistent with findings on neural suppression effects following repeated exposure to stimuli.

## 4. Sensory Substitution

In a pioneering study, Bach-y-Rita et al. [92] developed a haptic feedback device to convey information about the environment through vibrations to blind and blindfolded observers. The vibrations were applied to the observer’s back and following training their sensations changed to being felt as coming from the environment rather than the chair, but only if they actively trained with the device. Similarly, in Bach-Y-Rita et al. [93], participants reported externalization of the experiences following considerable training.

The metabolic rate of the visual cortex in the congenitally blind is as high as in sighted people. This may reflect a lack of neural pruning in early brain development and that the neurons are still active [53]. But the story is probably more interesting than this. Blind individuals receive information about the environment through other senses, but the brain maintains its normal functionality, since in 95% of cases, the visual impairment is caused by problems in the eye, retinae or visual pathways [60]. Sensory substitution offers the possibility of regaining lost perceptual abilities through neural plasticity, which allows “an adaptive response to a functional demand” [54].

Sadato et al. [55] showed, using positron emission tomography (PET), that Braille reading activated the primary and secondary visual cortex in blind people. To compare the effect of tactile stimulation both blind and sighted participants had to solve non-Braille tactile tasks. In the non-discrimination task participants swept their finger over a surface covered with Braille dots without responding. They either judged whether two grooves were parallel or not; whether two grooves were of the same width, or whether two of three uppercase letters (made of Braille dots) were identical. The discrimination tasks activated the primary visual cortex in the blind (although not to same extent as Braille reading). The opposite effect was observed in sighted people. These results show that the visual cortex in blind people processes non-visual stimuli.

In a study highly related to Sound of Vision, de Volder et al. [56] trained both early-blind (EB; all became blind within the first 3 years of life) and SCs in using an ultrasonic echolocation device to locate obstacles. The blind participants were able to detect a pole (width = 9 cm; height 200 cm) located 6 m away from them. The main purpose was to compare activity in the visual cortex in EBs and SCs. Before the study participants (SCs were blindfolded) practiced using the device in 6 one-hour lessons over a 3 to 4 week period; in recognizing a wall, a pole, doors and stairs while walking. To evaluate their proficiency of using the device, the subjects estimated the location of 2 poles and indicated which one was closer. In a comparison task the stimuli were tones. In the main experimental session, the task was to estimate the direction to, and distance of, a similar pole as was used during training. Using PET, de Volder et al. [56] found that the metabolic rate in the primary and associate visual cortex of the EBs was higher in both the control and the sensory substitution task than for the SCs. Furthermore, there was a trend within the EB-group for higher metabolic rates in the sensory substitution than the control task, a trend not observed for the SCs. Although the difference in the metabolic rate in the sensory substitution task and the control task was not quite significant, the results from de Volder et al. [56] can be considered evidence for sensory substitution.

Collignon et al. [57] applied repetitive transcranial magnetic stimulation (rTMS) to right dorsal extrastriate cortex. The signals interfered with auditory tasks performed by blind subjects but not for blindfolded sighted participants. Participants wore a head mounted video camera and images were transformed into sounds transmitted through headphones. When rTMS was applied, the performance of the EBs was significantly worse than without rTMS. In the auditory control task EBs made fewer errors when discriminating between intensity but neither group showed any effect of rTMS stimulation. This shows how the visual cortex of EBs is involved in auditory discrimination.

Amedi et al. [58] compared the patterns of activation generated by tactile object recognition tasks for eight sighted and congenitally blind subjects. The results demonstrated that object identification activated the LOC (lateral occipital complex) of the blind individuals in the same way as the sighted controls, providing evidence for the multisensory representation of visual images in the occipital cortex determined by alternate modalities, such as touch. The LOC is responsible for processing shape information as a result of three dimensional (3D) haptic perception [94], being responsive to tactile stimuli rather than auditory cues [95].

As discussed above, VIPs have advantages over sighted people in auditory processing and their visual cortex is more highly activated when auditory stimuli are processed compared to sighted people (see also [96,97]). In most of these studies the stimuli have been rather simple. Importantly, it also seems possible to train blind people to recognize body shapes and body posture using visual-to-auditory sensory substitution [59]. After ≈ 70 hour training (of which 10 were dedicated to body shape perception) 7 congenitally blind participants were able to correctly categorize the 2 dimensional images they were tested on, with an accuracy of ≈ 80% and some were even able to determine body posture. Striem-Amit and Amedi [59] used fMRI to demonstrate that the auditory cortex of the CBs was highly activated and the visual cortex (especially the right extrastriate visual-cortex) of the SCs as might have been expected. More interestingly, the right extrastriate visual cortex of the CBs showed the highest activation. This means that for congenitally blind and sighted controls the same “visual” brain areas were activated during the task [59].

The occipital cortex has mainly been thought to process visual information. But the results reviewed above demonstrate that the visual cortex is activated by input from other sensory modalities, such as hearing and touch. The brain is therefore task-oriented rather than sensory-specific, supporting the “task-machine” brain hypothesis [60].

In conclusion, there seems to be considerable agreement it the literature regarding preserved activity in visual brain areas in the blind and that there is often more activity in these areas in response to auditory and tactile stimulation in the blind than the sighted (see e.g., [98]). But the enhanced activity may not always lead to behavioral benefits for the visually impaired. Finally, note that for the purposes of sensory substitution, the capabilities of the sensory systems used to augment or replace visual perception have to be taken into account. For example, the bandwidth of information transmission in the visual system is higher than for both audition and tactile sensation [99]. We agree with Loomis et al, who argue that more basic research on this is needed that takes into account the spatial and temporal characteristics of the wide range of stimuli that must be conveyed with such a system.

## 5. Training Sound Localization

In order to decode the direction of a particular sound source in space, the human auditory system uses a set of binaural (for localization in the horizontal plane) and monaural cues (for localization in the median plane and front-back discrimination). The auditory system transforms the binaural and monaural cues into reliable spatial information. This is captured in the concept of the Head Related Transfer Function (HRTF) [100,101]. This process is probably influenced by associations between the visual and auditory systems that train and recalibrate sound localization [67]. Learning is a conscious and explicit change, where the observer is aware of the modifications occurring in the spatial auditory perception, while adaptation and spatial auditory recalibration indicate a complete, long-term and unconscious transformation that improves spatial resolution and sound localization performance [102].

Sound localization performance can be altered with ear blocks, by producing anatomical transformations to the shape of the pinnae, electronic hearing devices (hearing aids or cochlear implants) or the use of 3D sounds generated in virtual auditory environments [102].

Three main training methods have been used to enhance sound localization performance under altered listening conditions [102]. In the sound exposure paradigm participants unconsciously learn to adapt to altered auditory cues through multisensory feedback. The “training with feedback” paradigm involves a sound localization task that provides perceptual feedback concerning the correct direction of auditory stimuli in space. Thirdly, during “active learning” participants are actively involved in the process of spatial auditory adaptation, such as when participants are required to explore a virtual auditory environment by playing a game in which they identify the directions of various sound sources [74].

The simplest method for testing the human auditory system’s degree of adaptation to degraded auditory cues is based on inserting ears plugs in one ear [61,62,63,64,65]. Florentine [62], made participants wear a unilateral block from 5 to 101 days, Bauer et al. [61] inserted the ear block for 65 hours, van Wanrooij and van Opstal [103,104] produced a monaural spectral disruption for 9 to 49 days, while Held [66] applied an electronic hearing device that presented sounds with displaced azimuths of 20°, for 8 hours daily, constantly monitoring the degree of spatial auditory adaptation in specifically-designed periodical tests. With such methods, sound localization is initially poor, but recovers significantly over a period of several days. Held [66] had subjects record a displacement of the auditory representation halfway in the direction of the sound source’s azimuth shift [102].

In Hofman et al. [67], sighted observers were required to wear custom-made molds inside the concha of both ears for 6 weeks, which severely impaired sound localization accuracy in the vertical plane. However, several days after removal, auditory localization performance improved gradually and sound localization reached baseline. This shows that adult humans are able to adapt to altered spectral cues and acquire a new representation of pinna transfer functions for each listening condition (mold and no-mold) but also that participants developed two distinct spatial maps for the two conditions.

Carlile and Blackman [68] investigated the rate of adaptation to new spectral cues for 76 equally distributed sound sources located inside and outside the listener’s field of vision, applying small bilateral ear molds to the outer ear [69]. Immediately after insertion, localization was strongly affected while following 40.5 days of wearing the molds, front-back confusions, polar angle error and lateral (azimuth) error decreased significantly. Moreover, there was no difference between spatial localization performance recorded within the audio-visual field and other angular locations around the listener. This shows successful remapping of the spectral cues for localization and positional discrimination in the absence of visual stimuli. After the molds were removed spatial resolution returned to initial values (as in [67]), demonstrating that after exposure to altered listening conditions, the brain still preserves the representation of the “old” spectral cues.

For the “training with feedback” paradigm, observers receive explicit feedback on the accuracy of localization of the incoming sound source. For instance, Musicant and Butler [63] trained sound localization abilities in the horizontal plane. Observers received feedback about whether their choice was right or wrong. Strelnikov et al. [71,105] compared the degree of sound localization for three experimental groups: the first receiving only sound exposure, the second group received correct/incorrect feedback while the third group received perceptual feedback based training through visual and auditory cues about the correct direction of the sound source in space. Performance for this last group was the best followed by the group receiving correct/incorrect feedback.

### 5.1. Visual Feedback

In Carlile [69], four participant groups wore pieces of mold for 10 days. The control group received no feedback during training, the second group received only visual feedback through an LED, the third received audio and visual feedback, while the fourth received audio and visual training with the room lights turned on. The largest improvement was seen for the group that received visual and auditory feedback. In Zahorik et al. [70], the visual stimuli were a head orientation “crosshair” pointing directly ahead from the position of the listener’s head, a small point of light with high contrast that provided feedback about the correct direction of the sound source and a visual stimulus that indicated the front reference direction of 0° azimuth and elevation. In a post-test, 4 days after training, the rate of reversal errors decreased for 4 of the 6 listeners, from about 38% to 23%. In Strelnikov et al. [71], the rate of improvement was the smallest for the auditory training group, followed by the group receiving feedback about the correct response and finally, the highest for the audiovisual training group.

### 5.2. Proprioceptive Feedback

Honda et al. [72] divided participants into 2 groups: one that received proprioceptive feedback and a control group. Proprioceptive feedback decreased sound localization errors in the horizontal and in the vertical plane, regardless of head movement.

### 5.3. Haptic Feedback

In Bălan et al. [73] multimodal (haptic and auditory) based training was used, aimed at reducing the incidence of both reversal and azimuth errors in a virtual auditory environment where 3D sounds were synthesized using non-individualized HRTFs from the MIT database [106]. The stimuli consisted of a combination of white and pink noise in varying proportions, corresponding to the direction of the sound source in space and a narrowband “ding” sound, with 250 ms breaks between each burst. A haptic belt they wore on their heads contained 12 vibration motors, placed at 30° intervals around the head. Nine visually impaired adults (with residual vision from 0% to 20%) underwent a pre-test, training and a post-test measurement. During training, they listened to 12 auditory stimuli (for both types of sounds) and indicated their perceived direction using the hour hand of a clock. Consequently, the subjects felt a series of vibrations on the haptic belt, paired with the current stimulus in the headphones. Moreover, those who had a larger degree of residual vision could also receive visual feedback, as the correct direction of the sound was displayed in green along with the listener’s choice (in red). For the white-pink noise combination the mean rate of front-back confusions reduced from 12% to 6%, while the mean azimuth error decreased from 37° to 27°. Moreover, for the “ding” sound, the mean reversal error rates decreased from 14.3% to 12.5%, demonstrating rapid adaptation of visually impaired subjects to the perception of degraded sound cues through multimodal (haptic and auditory) feedback.

In “active training” paradigms [102] participants are actively engaged in improving their spatial auditory localization skills. The feedback they receive results from their own interactions with the environment. For instance, [74] trained 9 sighted participants on sound localization in a virtual auditory environment with audio and kinesthetic feedback. In the pre- and post-test sessions, the listeners indicated the perceived direction of a sound by pointing to it. During adaptation they searched for animal sounds hidden around them using a hand-held position track ball that provided kinesthetic and proprioceptive feedback. Both angular precision error and front-back confusion rates were reduced following training. The authors argued that sound localization improvements were triggered by adaptation of the human auditory system to the perception of 3D sounds. Moreover, this study demonstrates that visual information is not essential for improved sound localization, as it can be effectively replaced with, for example, kinesthetic, proprioceptive and vestibular feedback.

In Mendonça et al. [75] sound localization in the horizontal and vertical planes was tested. During training, participants identified the direction of incoming sound sources from 4 possible positions (0°, 30°, 60°, 90°) by pointing to their corresponding areas on a touch screen. Training continued until participants reached a rate of 80% correct for azimuth discrimination and 70% for elevation localization on two consecutive trial blocks. All participants improved their sound localization performance, even one month following training and notably, training was also helpful for untrained positions.

### 5.4. Auditory Stimulation (i): Free-Field

Honda et al. [72] used 36 loudspeakers, located at 0° and ± 30° intervals in the horizontal plane and at 0° and ± 30° in the vertical plane (12 speakers in each vertical row). Strelnikov et al. [71] used 15 loudspeakers arranged in the horizontal plane. For monaural listening following visual and auditory training, there was a reduction in the sound localization error in the horizontal plane, although accuracy did not reach levels obtained during binaural listening.

### 5.5. Auditory Stimulation (ii): Virtual Auditory Environments

In virtual auditory environments, the sounds are synthesized using recorded sets of HRTFs and transformed into 3D sounds delivered through headphones, in order to convey a high fidelity representation of the auditory scene, resembling free-field listening conditions. Recording HRTFs is time-consuming, the majority of virtual auditory displays therefore use generic sets of HRTFs that lead to localization errors and front-back confusions. Training is therefore necessary in order to familiarize observers with virtual auditory stimuli that use non-individualized localization cues. In Parseihian and Katz [74], the results for trained listeners were compared to the results of a control group who used individualized HRTFs. Significant improvements in sound localization accuracy were found, especially in the vertical plane. In Majdak et al. [107], the angular localization error decreased from 23° to 19° after training. Other experiments that used synthesized 3D sounds with non-individualized HRTFs are [75,108,109].

### 5.6. Head Dynamics

Honda et al. [72], used head-movement-dependent and head-movement-free listening conditions. Both unrestricted head movements and proprioceptive feedback led to decreases in sound localization errors in both the horizontal and vertical planes. Head movements during training are important for ensuring long-lasting localization improvement, especially in the horizontal plane. Additionally, head and body dynamics strongly influence spatial auditory perception, especially when sound duration exceeds 250 ms [110]. Modifications in binaural cues lead to a lower incidence of front-back confusions [111].

### 5.7. Auditory Stimuli

The sound stimuli used in the experiments discussed here were triangle waves [72], white noise [70,71,74,75,107] pink noise [108] or speech stimuli, convoluted with non-individualized HRTFs [75]. The best localization is typically seen for broadband sounds, such as clicks and noises, that have a more complex spectral profile and frequency variation [75,107].

### 5.8. The Effects of the Training Methods

In Mendonça et al. [75], training benefits still persisted after one month, suggesting that the human auditory system possesses the ability to consistently adapt to altered acoustic cues and that the learning generalizes to untrained locations and sounds. Moreover, the improvements seem to be long-lasting (even after 4 months) and do not alter the spatial map representation under real-world listening conditions, supporting Hofman et al.’s argument that training contributes to the development of a secondary spatial map in the brain [67,70]. In Mendonça et al. [75], improvements were visible one month following training, and also applied to untrained positions [105].

The results described in section 5 demonstrate that training can play an important role in spatial auditory recalibration to the perception of altered or artificial sound cues. Multimodal perceptual feedback-based training that combines auditory, visual, haptic or proprioceptive stimuli leads to significant improvements in localization in the horizontal and vertical planes and for spatial discrimination both in front and back. Some experiments indicate that visual information is not absolutely necessary for recalibrating spatial auditory resolution, and other forms of feedback can be effectively used for training sound localization performance of the blind or visually impaired. Moreover, in many cases, the effects of training proved to be long-lasting, demonstrating the reliability of multimodal feedback based training methods and high levels of adaptation acquired by the subjects.

## 6. Conclusions

Overall, the evidence that we have reviewed strongly indicates that it is feasible to assist visually impaired people by creating auditory representations of the surrounding environment and conveying it to them in real time, which is the goal of Sound of Vision. The demonstrated changes in brain function that can occur even in the less flexible adult brain, the adaptation effects and the sensory substitution effects we discussed all indicate that the brain is flexible enough for this to be viable. The evidence also suggests that mapping of provided auditory information to spatial codes may become less and less demanding with practice, so that strained connections can over time increase in fluency. The sound localization experiments described above demonstrate that training plays an important role in spatial auditory recalibration to the perception of altered sound cues and that these effects can be long lasting. In conclusion, long lasting changes following training such as that proposed in Sound of Vision may be expected, given the demonstrated flexibility of the cerebral cortex.

## Figures and Tables

**Table 1 brainsci-06-00020-t001:** Overview of the results reviewed in Section 2, presenting evidence for structural and functional differences in auditory and haptic perception between visually impaired and sighted people.

Paper	Main Results	Method
Park et al. [13]	Differences in cortical thickness, volume and area between VIPs and SCs.	MRI
Stevens and Weaver [14]	CBs show better TOJ and ABM performance than SCs.	Behavioral
Després et al. [15]	CBs are faster than SCs at locating a sound source from behind them in the vertical plain but not straight ahead. No significant differences in the horizontal plane.	Behavioral
Röder et al. [16]	In the peripheral condition N1 amplitude decreased faster for CBs than SCs indicating that CBs process peripheral auditory stimuli more efficiently than sighted people, but sighted people respond more accurately to central auditory stimuli.	Behavioral and EEG
Voss et al. [17]	No differnece between the performance of EBs and SCs when binaurally locating sound sources in the horizontal plane but a EB-subgroup performed worse in the vertical plane.	Behavioral
Finocdhietti et al. [18]	VIPs and SCs show similar performance in detecting the direction of a moving sound, but the performance of EBs in locating the end point of the moving sound is worse than for both LBs and SCs.	Behavioral
Fieger et al. [19]	SCs were slower in responding to peripheral than to central auditory stimuli. LBs were slower than the SCs but equally fast in responding to peripheral and central stimuli.	Behavioral
Fieger et al. [19]	CBs and LBs process peripheral auditory stimuli more efficiently than sighted people.	EEG
Lerens and Renier [20]	EBs respond faster to both central and peripheral auditory stimuli than Scs	Behavioral
Lessard et al. [21]	Found no differences in sound localization performance between congenital blind participants and sighted participants but the performance of their participants that had some residual sight was inferior compared to the other groups	Behavioral
Voss et al. [22,23]	No differences found between VIPs and SC, neither in locating sound sorces or comparing sound soureces locations	Behavioral
Gori et al. [24]	When judging whether the left or the right sound were closer to a central sound the CBs performed at chance level while the SCs solved the task adequately.	Behavioral
Gori et al. [24]	When comparing the distance of the first and third sound to the middle sound and when judging whether the sound was to the left or right of the base sound the performance of the CBs and SCs did not differ.	Behavioral
Röder et al. [25]	CBs responded significantly faster to acoustic stimuli than SCs.	Behavioral and EEG
Occelli et al. [26]	CBs showed lower JNDs than SCs in judging whether an auditory or tactile stimulus appeared first when stimuli appeared at different locations.	Behavioral
Röder et al. [27]	Better TOJ for CBs than LBs and SCs when comparing tactile stimuli.	Behavioral
Hötting and Röder [28]	When 3 or 4 tones are presented along with haptic stimuli the tones influence the performance less for CBs than SCs.	Behavioral
Hötting et al. [29,30]	CBs responded significantly faster to tactile stimuli than SCs but the difference was not significant for auditory stimuli.	Behavioral and EEG
Hötting et al. [29,30]	The initial selection process of simultaneously presented haptic and auditory stimuli may involve similar brain areas in CBs and SCs. The selection mechanism might however differ when stimuli have to be filtered by two factors (location and modality).	Behavioral and EEG
Wan et al. [31]	CBs and EBs, but not LBs, show better pitch and timbre discrimination than matched SCs.	Behavioral
Postma et al. [32]	Both EBs and LBs were faster than sighted blindfolded people (SB) in putting geometric shapes in their correct places on a wooden board. After rotating the board 90° the VIPs were still faster than the Sbs	Behavioral
Withagen et al. [33]	VIPs faster than SCs in haptically comparing artificial objects and telling whether they were the same or not.	Behavioral
Goldreich and Kanics [34]	VIPs show better tactile acuity than SCs.	Behavioral

**Table 2 brainsci-06-00020-t002:** Overview of the results reviewed in Section 3 (neural adaptation).

Paper	Main Results	Method
Maljkovic and Nakayama [35]; Kristjánsson and Jóhannesson [36].	Repeated presentation of stimulus properties, e.g., color and shape reduces RTs.	Behavioral
Maljkovic and Nakayama [35]; Ásgeirssson et al. [37].	Repeated presentation of the stimuli at the same location reduces RTs.	Behavioral
Dehane et al. [38]; Kristjansson et al. [39], Grill-Spector and Malach [40]; Kourtzi and Kanwisher [41]; Larsson and Smith [42].	When particular stimuli repeat, this results in suppression of the BOLD signal as measured with fMRI.	fMRI
Krekelberg, Boynton and van Wezel [43].	Repetition suppression may cause selective “pruning” of the neuronal population response in a given situation.	Review on fMRI
Kovács et al. [44]; Vizioli et al. [45]	Adaptation, in general, reduces EEG signals	EEG
Rentzeperis et al. [46]	When stimulus’ color was repeated the amplitude of the N1 component became significantly lower than when color did not repeat.	EEG
Rossion and Boremanse [47]	When the same face was repeatedly presented, amplitude measured over the occipito-temporal scalp decreased significantly compared to when the face on the current trial differed from the previous one.	EEG
Gerlicher et al. [48]	Adaptation occurs for repeated faces with neutral emotional expressions but not threatening expressions.	EEG
Dobbins et al. [49]	RTs are faster for previously presented, versus novel stimuli, in conjunction with BOLD repetition suppression in a number of brain regions.	Behavioral and fMRI
Wander et al. [50]	When moving a cursor over the screen through BCI, activity as measured with ECoG (implanted electrodes) in a number of brain areas decreased as proficiency on the task increased	BCI ana ECoG
Bichot and Schall [51,52]	Single cell recording in frontal eyefields of Macaque monkeys showed decreased neuronal activity when stimulus properties were repeated	Single cell recording

Studies presented in the table were all conducted with human participants except the study of Bichot and Schall [51,52] performed on Macaque monkeys.

**Table 3 brainsci-06-00020-t003:** Overview of the results reviewed in Section 4, Sensory substitution.

Paper	Main Results	Method
De Volder et al. [53]	The metabolic rate of the visual cortex in the congenitally blind is as high as in sighted people indicating that it’s neurons are still active.	PET and MRI
Bach-y-Rita and Karcel [54]	Sensory substitution may offer the possibility of regaining lost perceptual abilities through the process of plasticity.	Review on sensory substitution
Sadato et al. [55]	Braille reading activated the primary and secondary visual cortex in blind people.	PET
De Volder et al. [56]	When an ultrasonic echolocation device was used to locate obstacles PET studies revealed that the metabolic rate in the primary and associate visual cortex of EBs was higher than for SCs.	PET
Collignon et al. [57]	rTMS of visual cortex interfered with auditory tasks performed by blind subjects but not for blindfolded sighted participants. Shows that the visual cortex of EBs is involved in auditory discrimination.	rTMS
Amedi et al. [58]	LOC of blind individuals was activated in the same way as the sighted controls during an object-localization task.	fMRI
Striem-Amit and Amedi [59]	When using a visual-to-auditory sensory-substitution device to recognize body shapes and body posture fMRI revealed that the right extrastriate visual cortex was highly activated during the task in both CBs and SCs.	fMRI
Maidenbaum et al. [60]	The brain is task-oriented rather than sensory-specific, supporting the “task-machine” brain hypothesis.	Review on sensory substituion

**Table 4 brainsci-06-00020-t004:** Overview of the results reviewed in Section 5 (training sound localization).

Paper	Main Results
Bauer et al. [61]; Florentine [62]; Musicant and Butler [63]; Slattery and Middlebrooks [64]; Kumpik et al. [65].	Low initial sound localization accuracy, but significant improvements after a period of several days.
Held [66]	Displacement of auditory representation halfway in the direction of the sound source’s azimuth shift. Localization accuracy improved after a few days.
Hofman et al. [67]; Carlile and Blackman [68]	Immediately after mold insertion, localization was severely impaired, but after several days performance improved gradually, reaching pre-training accuracy levels when the molds were removed.
Carlile [69]; Zahorik et al. [70]; Strelnikov et al. [71].	Improvements in sound localization, especially decreased front-back confusion.
Honda et al. [72].	Decreases in sound localization errors in the horizontal and in the vertical plane, regardless of head movement
Bălan et al. [73]	Decreases in sound localization error and front-back confusion rate for broadband and narrowband sounds.
Parseihian and Katz [74]	Reduced angular precision errors and front-back confusion
Mendonça et al. [75]	Sound localization performance improved for all participants

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
