# Peer review of "The Sound of Vision Project: On the Feasibility of an Audio-Haptic Representation of the Environment, for the Visually Impaired"

_brainsci, 2016, doi:10.3390/brainsci6030020_

Round 1

Reviewer 1 Report

This was a difficult paper to evaluate.  On the positive side, it was well-written and each of the sections was a good review of the sub-field.  The main problem - which is a fundamental one - is that it isn't clear what the paper is actually about!  The title and abstract imply that the paper concerns the feasibility of 'the sound of vision project', but actually we are given no information about that project in order to form a judgment of it.  For instance, what kind of visual information would be represented (colour, depth, luminance, etc.) and how (using what auditory and tactile properties)?  What would be the function of the device?  Instead, the paper reads more as a list of principles of brain reorganisation following blindness.  This is interesting but is done elsewhere (e.g. see the 2014 Special Issue of Neuroscience and Biobehavioral Reviews).  Also, the sections do not hand together very well as a whole and need to be carefully linked together around an over-arching theme.  To give one example, it wasn't clear what the purpose of the section on 'neural adaptation' was.  It referred to very broad principles concerning priming etc. that seemed tangentially related to the main there.

In short, the authors need to recast the paper in terms of a specific proposal or as a more general review.  If the latter, then it needs to be clear about what the new focus of the review is. 

Author Response

 Reviewer: This was a difficult paper to evaluate.  On the positive side, it was well-written and each of the sections was a good review of the sub-field.  The main problem - which is a fundamental one - is that it isn't clear what the paper is actually about!  The title and abstract imply that the paper concerns the feasibility of 'the sound of vision project', but actually we are given no information about that project in order to form a judgment of it.  For instance, what kind of visual information would be represented (colour, depth, luminance, etc.) and how (using what auditory and tactile properties)?  What would be the function of the device?  Instead, the paper reads more as a list of principles of brain reorganisation following blindness.  This is interesting but is done elsewhere (e.g. see the 2014 Special Issue of Neuroscience and Biobehavioral Reviews).  Also, the sections do not hand together very well as a whole and need to be carefully linked together around an over-arching theme.  To give one example, it wasn't clear what the purpose of the section on 'neural adaptation' was.  It referred to very broad principles concerning priming etc. that seemed tangentially related to the main there.

In short, the authors need to recast the paper in terms of a specific proposal or as a more general review.  If the latter, then it needs to be clear about what the new focus of the review is.

Our response: We are grateful to the reviewer for the positive comments about the manuscript, while we also take the reviewers more critical comments to heart and in the revision we have tried to make our goals with manuscript clearer. To briefly summarize, our specific proposal is that the evidence indicates that sensory substitution for blindness is overall feasible, but also that there are various pitfalls that must be avoided and the Sound of Vision (SoV) project will try to sidestep. We discuss these pitfalls in the manuscript. Additionally, we have provided more info on the specific aims of SoV. Our overview is organized as follows. We use the first section to review examples of neuroplasticity of the adult brain specifically emphasizing functional changes in the visually impaired. In section two we discuss adaptation, specifically long-term and short-term effects of repeated exposure to particular stimuli on brain activity. We feel that this question is very important since it touches on the issue of how we may expect users to adjust to any sensory substitution equipment. In section three sensory substitution is discussed such as when a region devoted to a particular function takes on another function (see overview in Table 2). In section four we discuss research on adaptation to changes in auditory environments (see Table 3). In section five we review evidence pertaining to the training of sound localization (see Table 4 for overview). Finally, we present concluding remarks upon our review of neuroplasticity, adaptation and sensory substitution and the implications for the feasibility of the Sound of Vision sensory substitution device.

We have taken to heart the reviewer’s criticisms regarding the specific aims and also the relevance of each section to the theme of the paper, and have tried to make this argument flow more naturally in the revised manuscript. We have also tried specifically to clarify the relevance of the section on adaptation. To our minds, adaptation is important regarding sensory substitution simply because it informs us of how our systems respond to long exposure. And note that we talk about the connection between short-term and long-term adaptation in this context (which is very important, to our mind).

We are quite certain that the questions that we raise have not been addressed before in the same way in other publications, although we acknowledge that there is, probably unavoidably, considerable overlap with the discussion in other papers.

While we feel that we done our best to address the criticisms of the reviewer we emphasize that we would be happy to make other changes that may become necessary before the manuscript can be published.

Finally, we thank the reviewer for his constructive criticisms and many excellent suggestions, and we note that we will happily make other changes that may be deemed necessary before the manuscript can be published.

Reviewer 2 Report

The research article from Johannesson et al. reviews the neuroplasticity of the brain in visually impaired people. The article is interesting, quite well written, but still not complete and difficult to read in the present state.

1.      Introduction

I won't compare the simplicity of using a mouse to a sensory substitution device (SSD). SSDs are quite complex to use in everyday life (see a review from  Elli et al. Multisensory Research, 2014).

To jump the gun, is an idioms. You should replace it with a more appropriate expression.

Line 68: you should provide newer references, for example from the work of Andrew King,Oxford University.

2. Evidence for structural and functional differences in auditory....

I suggest to shorten the description of each study and to include additional pivotal studies as Gougoux et al 2005, Collignon et al. 2009, Lessard 1998, Voss et al. 2004, Lewald 2002, Finocchietti et al. 2015, Voss et al. 2015. More details could be added in the table, adding at least a column indicating the methodology used. The table is also difficult to read, as there isn't any line separating the main results of each study presented.

3. Neural Adaptation

I would shorten the description of each study and add a column in table 2. Some of the studies presented are on human, some others are on animals. They should then properly divided in the table.

4. Sensory substitution

I would shorten the description of each study and add a column in table 3. I would add some of the early studies from Bach Y Rita, who pioneered the SSDs.

5. Training sound localization

I would divide table 4 in relation to the feedback used.

Overall:

As caption for each table, instead of writing “overview of the results reviewed in section N”, I would write  “overview of the results reviewed in the neural adaptation section” for example. It reminds what the section is about.

Abbreviations should be all presented just at the end, they are not needed for each table.

There is a weird numbering of pages (for example 11 of 4?).

Author Response

The research article from Johannesson et al. reviews the neuroplasticity of the brain in visually impaired people. The article is interesting, quite well written, but still not complete and difficult to read in the present state.

Our response. We thank the reviewer for his positive view on our manuscript and all the highly useful suggestions which we believe will improve our manuscript. We have done our best to respond adequately to the reviewer’s comments but will be happy to make further changes that may be deemed necessary for publication.

1.      Introduction

I won't compare the simplicity of using a mouse to a sensory substitution device (SSD). SSDs are quite complex to use in everyday life (see a review from Elli et al. Multisensory Research, 2014).

Our response. We agree that the SSDs are still rather complicated to use, especially compared to the computer mouse. But the computer mouse was not always as easy and straightforward to use as it is now, and it required REMAPPING. We do not wish to equate learning to use a mouse with learning to use a sensory substitution device, but we feel that the remapping involved in learning to use a computer mouse use is a good intuitive example of the remapping problems that design of sensory substitution devices entails. The example demonstrates the brain’s capacity to recalibrate its operative characteristics to meet new demands. We therefore actually feel that it is quite pertinent to questions of sensory substitution. Furthermore, the goal of Sound of Vision is to develop a system that will be relatively easy to use and we believe that all other working on similar project have the same goal. We also believe that, maybe not at such a distant moment in the future, that SSDs will be in common use (as the mouse is now) among people who need them and that the SSDs will be increasingly easy to use because of peoples’ familiarity with the principles involved. We, therefore, respectfully, disagree with the reviewer and prefer to keep this paragraph as it is. However, we are certainly ready to reconsider our decision if deemed necessary for publication.

To jump the gun, is an idioms. You should replace it with a more appropriate expression.

Our response: We have changed this.

Line 68: you should provide newer references, for example from the work of Andrew King,Oxford University.

Our response. We thank the reviewer for this suggestion and we found some interesting papers on this and added 4 of them to our reference list [2,3,4,5].

2. Evidence for structural and functional differences in auditory....

I suggest to shorten the description of each study

Our response. We have shortened the description of the studies as much as we consider reasonable but will be happy to try to shorten them more if deemed necessary for publication.

and to include additional pivotal studies as Gougoux et al 2005, Collignon et al. 2009, Lessard 1998, Voss et al. 2004, Lewald 2002, Finocchietti et al. 2015, Voss et al. 2015.

Our response. We thank the reviewer for pointing out these interesting studies and we now mention most of them [21, 22, 23, 24] in the manuscript and hope that this is sufficient (although covering every single relevant reference will always be impossible). We would, however, be happy to include more references if necessary for the publication of our manuscript, but again note that our citations of the literature can never be exhaustive.

More details could be added in the table, adding at least a column indicating the methodology used.

Our response. We have added columns to the tables where the methodology in each study is briefly explained.

The table is also difficult to read, as there isn't any line separating the main results of each study presented.

 Our response. We agree with the reviewer. We have now added dotted lines between the studies in the table. Furthermore, we believe that the centering of the text (formatted by the journal) made the table harder to read and we have now justified the text in the cells. We hope these changes will make the table easier to read. We have made the same formatting changes to all the other tables.

3. Neural Adaptation

I would shorten the description of each study

Our response. We have shortened the description of the studies as much as we consider reasonable but will be happy to try to shorten them more if deemed necessary for publication.

and add a column in table 2.

Our response. We have now added one column with the method used in each study.

Some of the studies presented are on human, some others are on animals. They should then properly divided in the table.

Our response. This is an important suggestion and we thank the reviewer for it. We have now sorted the table so the paper including animal research is the last one in the table and we point it out in the table’s caption.

4. Sensory substitution

I would shorten the description of each study

Our response. We have done our best to shorten the descriptions but would be happy to try to shorten them further if that might contribute to publication of our manuscript.

and add a column in table 3.

Our response. We have now added one column with the method used in each study.

I would add some of the early studies from Bach Y Rita, who pioneered the SSDs.

Our response. We do agree with the reviewer; we should have done that. Now we open the discussion of sensory substitution by referring to two pivotal studies from the Bach-y-Rita research group [68, 69].

5. Training sound localization

I would divide table 4 in relation to the feedback used.

 Our response. In table 4, our main emphasis is on the main results from each study and we like to keep studies with similar results in the same cells. To be able to order the table as the reviewer requests, we would need to distribute these studies between several lines. Although that the reviewer’s suggestion is good, we believe that the table conveys its message better as it is now. However, we will certainly reconsider this if deemed necessary for publication

Overall:

As caption for each table, instead of writing “overview of the results reviewed in section N”, I would write “overview of the results reviewed in the neural adaptation section” for example. It reminds what the section is about.

Our response. We believe this is a very good idea and have changed the table captions accordingly.

Abbreviations should be all presented just at the end, they are not needed for each table.

Our response. This is a fine suggestion, and we have now removed the abbreviations from the tables, and they can be found at the end of the manuscript.

There is a weird numbering of pages (for example 11 of 4?).

Our response. We thank the reviewer for pointing this out and we have now fixed this.

Finally, we thank the reviewer for his constructive criticisms and many excellent suggestions, and we note that we will happily make other changes that may be deemed necessary before the manuscript can be published.

Round 2

Reviewer 1 Report

My previous comment has not been properly addressed...

"The title and abstract imply that the paper concerns the feasibility of ʹthe sound of vision
projectʹ, but actually we are given no information about that project in order to 
form a judgment of it. For instance, what kind of visual information would be 
represented (colour, depth, luminance, etc.) and how (using what auditory and 
tactile properties)?"

Basically the paper is about the feasibility of sensory substitution as a general enterprise (of which 'sound of vision' is one example) rather than the feasibility of their own project in particular (the specific details of which are not available for us to evaluate).  If the authors are willing to make this point very clear then I would be willing to recommend acceptance. 

Author Response

Reviewer 1

"The title and abstract imply that the paper concerns the feasibility of ʹthe sound of vision
projectʹ, but actually we are given no information about that project in order to 
form a judgment of it. For instance, what kind of visual information would be 
represented (colour, depth, luminance, etc.) and how (using what auditory and 
tactile properties)?"

Our response. We do not aim to provide information about color or luminance and similar qualities of the environment and have considerable doubts that is it possible for now, without saying that it will never be possible. We aim for providing some kind of depth information but it will mostly be limited to the distance to the objects and – hopefully – the depth of the objects. For now, in the revised manuscript we provide a short description of what information we are going to represent and explain how this information will be conveyed to the user. We hope this is in accordance with what the reviewer is asking for.

Basically the paper is about the feasibility of sensory substitution as a general enterprise (of which 'sound of vision' is one example) rather than the feasibility of their own project in particular (the specific details of which are not available for us to evaluate).  If the authors are willing to make this point very clear then I would be willing to recommend acceptance. 

Our response. This is indeed what we wish to discuss in the manuscript. The issues of feasibility of sensory substitution that we discuss are by no means specific to Sound of Vision. We now try to make this point clear in the revised manuscript.

Finally, we thank the reviewer for his constructive criticisms and we note that we will happily make other changes that may be deemed necessary before the manuscript can be published.

Reviewer 2 Report

The research article is definitely improved in the last version. I still don't completely agree with the comparison between a mouse and an SSD, but I believe everyone is entitled to address his point of view.

I still have a minor revision to ask: besides the article by Gori et al., nowadays it is widely accepted a trade-off in the auditory localization abilities of CG subjects. see Voss et al. 2015, King 2015 (it is a review), plus other studies from Gori's group (Finocchietti et al. 2015, Cappagli et al. 2015). This should be addressed in the manuscript.

Author Response

Reviewer 2

The research article is definitely improved in the last version. I still don't completely agree with the comparison between a mouse and an SSD, but I believe everyone is entitled to address his point of view.

Our response. We thank the reviewer for his positive view of what we did to improve our manuscript. We are also happy about the reviewer’s opinion concerning the mouse and the SSD. So maybe we can agree to disagree on that point? It is not a major issue for us to include this point, but we nevertheless do like this analogy.

I still have a minor revision to ask: besides the article by Gori et al., nowadays it is widely accepted a trade-off in the auditory localization abilities of CG subjects. see Voss et al. 2015, King 2015 (it is a review), plus other studies from Gori's group (Finocchietti et al. 2015, Cappagli et al. 2015). This should be addressed in the manuscript.

Our response. We thank the reviewer this comment, which we believe is important to take into account when comparing the performance of visually impaired and normally sighted people. We now address this aspect in the revised manuscript by reviewing some of the papers the reviewer pointed out. We hope these additions meet the reviewer’s requirements but will be happy to improve them if this is deemed necessary for publication of our manuscript.
